



# Introduction of the Virtual Center of Wind Pressure for correlating large-scale turbulent structures and wind turbine loads

Carsten Schubert[1], Daniela Moreno[2], Jörg Schwarte[3], Jan Friedrich[2], Matthias Wächter[2], Gritt Pokriefke[3], Günter Radons[1,4,†], and Joachim Peinke[2]

[1]ICM – Institute for Mechanical and Industrial Engineering Chemnitz, Germany
[2]ForWind – Institute of Physics, University of Oldenburg, Germany
[3]Nordex Energy SE & Co. KG, Germany
[4]Institute of Physics, Chemnitz University of Technology, Germany
[†]deceased, 20 July 2024

**Correspondence:** C. Schubert (C.Schubert@icm-chemnitz.de), M. Wächter (matthias.waechter@uni-oldenburg.de)

**Abstract.** For modern wind turbines, the effects of inflow wind fluctuations on the loads are becoming increasingly critical. Using field measurements of a full-scale operating wind turbine and simulated loads calculated with reconstructed wind fields from wind measurements from the GROWIAN campaign, we identify particular load events that lead to high values of the so-called damage equivalent loads. Remarkably, the simulations do not reproduce such load occurrences when standard syn-
thetic turbulent wind fields are used as inflow. These standard wind fields are typically parameterized by statistics at a single measurement location (e. g., mean wind speed and turbulence intensity). In this article, we introduce a new characteristic of a wind field: the *virtual center of wind pressure*. The new feature is calculated from averages of the thrust force acting on a defined area, i. e., the rotor area of the turbine. We correlate these characteristics to the unusual load events observed in the operational measured data. Furthermore, we demonstrate that the introduced concept is an efficient tool to characterize large-
scale structures within wind fields. We propose using the virtual center of wind pressure in conjunction with the well-defined single-location properties to consolidate improved descriptions of atmospheric wind and more accurate wind fields for turbine simulations.

## 1 Introduction

The correct estimation of operational loads is of central importance for the design and certification of wind turbines. Current challenges and recent developments in wind energy have led to increased sized and slender turbines (Veers, 2019, 2023). As a result, the loads estimated with standard tools used for previous (i. e. smaller and stiffer) turbine designs are less and less comparable to measured loads. In particular, the accurate determination of loads caused by the turbulent inflow conditions is a





significant challenge due to not-yet-investigated interactions between larger elements of the turbine (i. e., turbine blades) and
larger scales and heights of atmospheric wind structures (Kuik, 2016; Veers, 2019).

To cover the broad range of operating scenarios over the lifetime of a wind turbine, the atmospheric conditions are usually specified by the International Electrotechnical Commission (IEC) guidelines (IEC, 2019). According to the IEC standard, turbulent wind fields are generated using the Kaimal (Kaimal, 1972) or Mann (Mann, 1998) models. These models are parametrized by simple statistical quantities, e. g., mean wind speed and shear, turbulence intensity, and integral or coherence length scales.
Here, simplified assumptions are made to describe the coherence of the wind field over a plane perpendicular to the main direction (Davenport, 1961; Thedin, 2022). The decrease of correlation with distance is typically assumed to be an exponential function. Moreover, extrapolations using power or logarithmic laws assume the change of wind speed with height in the so-called wind profiles. The exponents that parameterize these laws are typically fixed for a given location. However, analyses of wind measurements have proven the high variability of such velocity profiles, with even negative gradients (Gualtieri ,
2016; Wagner, 2011). Moreover, the wind profiles are calculated based on 10-min averages over height. Consequently, wind gradients on scales below 10-min and multiple gradients co-occurring at different horizontal locations at the plane are unresolved. Furthermore, the standard wind models do not account for extreme operating conditions, e. g., wind gusts and extreme shear. Therefore, extreme events are separately added by the IEC guideline. Particularly for wind gusts, the isolated event is assumed to have a Gaussian temporal evolution and be coherent over the whole rotor plane. However, asymmetry and non-
Gaussianity have been demonstrated (Hu, 2018). Consequently, modifications of such Gaussian and coherent assumptions have been proposed by Bierbooms (2024) for improving the validity of the IEC models.

Within the wind industry, the Blade Element Momentum (BEM) method is the prevalent approach for the simulation of loads. A full-scale wind turbine model is exposed to turbulent inflow within the BEM simulation with defined characteristics. Ideally, the loads calculated using these aero-servo-elastic simulations should accurately reproduce the loads experienced by
operating wind turbines, at least statistically. This implies that the range and distribution of the simulated loads for specific inflow conditions should align with measurements obtained during comparable operating circumstances. However, this alignment is not always achieved. Turbine manufacturers and operators report discrepancies of the loads between simulations and measurements. One possible explanation for these dissimilarities might be attributed to inaccuracies within the wind fields used for the numerical estimations. Unpredicted loads at operating turbines may be induced by structures occurring in the
atmosphere that have not been adequately included in the current models for generating synthetic wind fields within the IEC guidelines. Recently, different extensions of IEC wind field models have been suggested, focusing on empirically measured large-scale anisotropies in the marine boundary layer (Syed, 2024) and small-scale extremes (Friedrich, 2021, 2022; Yassin, 2023).

In this article, we introduce a new characteristic of the wind field termed the virtual Center of Wind Pressure (CoWP), which
demonstrates clear correlations to particular loads at the wind turbine. This concept offers two primary contributions. Firstly, it extends the current standard characterization of wind fields. Secondly, it proposes a potential tool for load estimations. The concept of the virtual center of wind pressure is based on the notion of the center of pressure widely used in fluid mechanics (Anderson, 1991). This quantity indicates the point-wise location of a theoretical aggregated version of the pressure field





acting on a body. In comparison to standard aggregated values in the wind energy context, such as the rotor-equivalent wind
speed (Wagner, 2010), which is solely a function of vertical displacement, the CoWP also accounts for horizontal "excursions"
of the wind field in the rotor plane. It should be noted that rotor-equivalent and sector-averaged wind speeds have mainly been
employed in the context of power output and load surrogate modeling for control purposes (Guilloré , 2024; Coquelet , 2024).
Nonetheless, a clear association with the occurrence of particular turbine loads remains to be established. As it will be demon-
strated throughout this paper, the dynamics of the CoWP are strongly correlated with the dynamics of tilt moments determined
from BEM simulations of the corresponding wind field. It can thus be used as a rough estimation method of potential load
characteristics directly aggregated from the wind field. For the definition and validation of the concept, three different data sets
are investigated: First, wind and load measurements from a full-scale operating wind turbine. Second, measured wind data by
the met mast array of the GROWIAN campaign (Koerber, 1988; Günther, 1998). Third, IEC standard synthetic wind fields
with their corresponding BEM estimated loads.

The paper is organized as follows: Sec. 2 introduces relevant definitions and methods and describes the data. In Sec. 3, the
motivation for the study is outlined, with the discrepancies between simulations and measurements being demonstrated. Next,
in Sec. 4, we investigate the cause of these differences. The main contribution of this paper is presented in Sec. 5 with the
introduction of the virtual center of wind pressure and its correlation to loads at the wind turbine. Finally, in Sec. 6, we present
the findings of our investigation and include some remarks on potential future work.

## 2 Definitions, Methods, and Data

In this section, we introduce the damage equivalent load, a well-known load estimator relevant to our analysis. Moreover, we
provide a short description of the numerical method for the load estimations and details of the data to be used as the basis of
our work.

### 2.1 Damage Equivalent Loads

A common criterion in the wind industry to predict the service life of the mechanical components of turbines is the so-called
*damage equivalent load* (DEL) approach (IEC, 2019; Sutherland, 1999). The DEL is a scalar quantity that quantifies the
damage induced by a one-dimensional load over a certain time span. By definition, it is a weighted sum of the amplitudes $s_i$
of the hysteresis cycles, where each amplitude is weighted with an exponent $m$. The DEL is then calculated as,

$$\text{DEL} = \left( \frac{\sum_{i=1}^{n} n_i s_i^m}{n_{ref}} \right)^{\frac{1}{m}} \quad , \tag{1}$$

where $n_i$ is the number of cycles with amplitude $s_i$, and $n_{ref}$ is a reference number of cycles, typically assumed as the number
of cycles to failure. The exponent $m$, known as the Wöhler exponent, is characteristic of the material and estimated from
the so-called S-N curves (Orowan, 1939). Values of $m \approx 4$ are used for welded materials, e. g., the tower or bearings, while
$m > 10$ are typical for fiberglass composite materials, e. g., the blades. According to Eq. 1, the larger the value of $m$, the
more dominant the largest amplitudes $s_i$ in calculating the DEL. More details on the DEL method and the Wöhler exponents



are found in (Sutherland, 1999; Minner, 1945). A comparison of the DELs between simulated and measured loads will be discussed in Sec. 3.

## 2.2 BEM Aeroelastic Simulations

In this study, we use the aero-servo-elastic simulation tool alaska/Wind (ICM, 2023), which is based on a general purpose multibody dynamics modelling system. The comparability of alaska/wind to other state-of-the-art aeroelastic simulation tools was demonstrated in Zierath (2016) and Hach (2020).

To cover geometrical nonlinearities, the structural blade models are represented directly by finite beam elements (Schubert, 2017). A Beddoes-Leishman-like dynamic stall model covers the unsteady aerodynamics, whereas a Dynamic Flex wake model (Hansen, 2008) covers the wake effects.

Further degrees of freedom used in the turbine model are: The drive train incorporates a radial degree of freedom to account for flexibility and a torsional degree of freedom modeling the gearbox. The yaw drive contains a nodding degree of freedom. The tower is modeled by a linearized flexible body containing side-side, fore-aft, and torsional degrees of freedom. The foundation is connected to the soil by soil springs. A generalized-alpha method is used to solve the system of equations of motion.

## 2.3 Wind and Load Data

The three data sets investigated in this work are now described:

(i) The first data set corresponds to operational data provided by Nordex Energy. 10-min measurements of both loads of the full-scale operating Nordex turbine and wind speed data from a met mast at the turbine's location are provided. The hub of the turbine is located at 125 m and the rotor diameter is 149 m. Wind measurements with a sampling frequency of 1 Hz at three different heights are available: at hub height at 125 m; at the lowest passage of the blade tip at 50 m, and at a height in between at 88 m.

The operational load measurements to be investigated were selected by the manufacturer. The aim was to collect data spanning a wide range of wind conditions, i. e., mean wind speed and turbulence intensity, with enough occurrences of specific load events that will be discussed in Sec. 3. The load measurements have a frequency of 50 Hz. Due to confidentiality, the measured loads are normalized by a scaling factor.

(ii) The second data set corresponds to synthetic wind data and the resulting loads calculated through BEM simulations. Synthetic turbulent wind fields are generated by the IEC-standard Kaimal wind model (Kaimal, 1972). The 10-min turbulent fields aim to mimic the characteristics of the measured atmospheric data provided by Nordex. More details of the specific parameters for generating the synthetic fields are given in Sec. 3. The loads on the turbine resulting from these Kaimal wind fields are calculated with the alaska/Wind simulator described in Sec. 2.2.

(iii) The third data set corresponds to wind measurements from the GROWIAN campaign. The data was recorded at the site of the 3 MW wind turbine GROWIAN project near the German coastline at the North Sea. Two met-masts with





measurement devices at five different heights (from $50\,\mathrm{m}$ to $150\,\mathrm{m}$) covered an area of $76 \times 150\,\mathrm{m}^2$. The wind speed data were sampled at $2.5\,\mathrm{Hz}$. A total of 334 collections of time series (i. e., simultaneous from all the anemometers) are available for investigation. More details about the measurement campaign are in (Koerber, 1988) and (Günther, 1998).

## 3 Damage Equivalent Loads: Differences between measured and simulated data

The main motivation of our research is to improve our understanding of the differences between the simulated (e. g., using Kaimal wind fields) and the measured loads at the full-scale Nordex turbine. In the following, we demonstrate such discrepancies between the simulated and the measured data. Specifically, we investigate bending moments at the main shaft of the wind turbine, i. e., the tilt and the yaw moments. For simplicity, we refer to the yaw and tilt moments at the main shaft as $T_{yaw}$ and $T_{tilt}$. Moreover, we will use the subscripts -$m$ and -$s$ to refer to measured loads and simulated loads, respectively. Two

subscripts thus identify the moments. For example, the measured yaw moment is noted as $T_{m,yaw}$, while $T_{s,tilt}$ indicates the simulated tilt moment.

The starting point of the comparison between measurements and simulations is based on the DEL defined in Sec. 2.1. Due to the significant potential effects on the turbine, particular interest is given to large-amplitude events within the load signals. To give predominance to such large amplitude events within the calculation of the DELs, a Wöhler exponent $m = 10$ is used

for the analysis (see Eq. 1). Then, the DELs will be referred to as DEL10.

Next, the DEL10 are conditioned by the simultaneous wind conditions. Each DEL10 calculated from a 10-min load signal is then classified into so-called wind bins. The wind bins are defined by the characteristics of the wind speed time series $u(t)$, i. e., the mean ($\bar{u}$) and the turbulence intensity (TI). The turbulence intensity is calculated as $\mathrm{TI} = \bar{u}/\sigma_u$, where $\sigma_u$ is the standard deviation. Both quantities, $\bar{u}$ and TI are calculated at hub height over the individual 10-min periods. Equally spaced wind bins

with a size of $\Delta\bar{u} = 1\,\mathrm{m/s}$ and $\Delta\mathrm{TI} = 2\,\%$ are defined, e. g., $\bar{u} = 7.0 \pm 0.5\,\mathrm{m/s}$ and $\mathrm{TI} = 9 \pm 1\%$.

Changes in the wind conditions are expected to be reflected in the calculated DEL10. To prove this, we evaluate the DEL10 from simulated data for different combinations of $\bar{u}$ and TI. We consider nine $\bar{u}$-TI combinations within the wind bin of $\bar{u} = 8 \pm 0.5\,\mathrm{m/s}$ and $\mathrm{TI} = 10 \pm 1\%$. Eight realizations of IEC-Kaimal turbulent fields with each $\bar{u}$-TI combination are generated. The conditions of $\bar{u}$ and TI are imposed at the height of $125\,\mathrm{m}$. The effect on the loads of the turbine is simulated via the

alaska/Wind BEM simulation. Fig. 1 shows the resulting DEL10 of the simulated $T_{s,tilt}$ for the different wind conditions. The horizontal green and red lines depict the global minimum, $\mathrm{DEL10} = 0.087$, and global maximum, $\mathrm{DEL10} = 0.154$, over the 72 realizations of the wind bin.

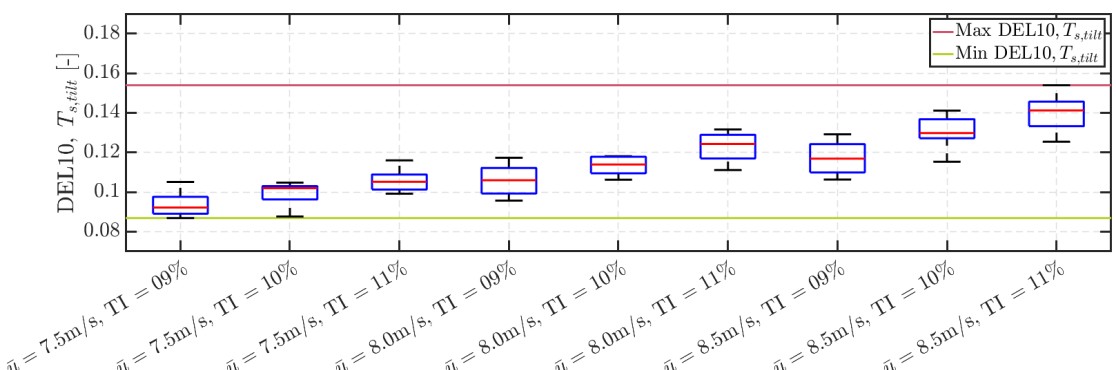

**Figure 1.** DEL10 of the simulated tilt moment ($T_{s,tilt}$) at the main shaft of the model wind turbine. Nine $\bar{u}$-TI combinations are evaluated within the wind bin $\bar{u} = 8 \pm 0.5$ m/s and TI= $10 \pm 1$%1. The global minimal and maximal values of DEL10 are marked by the green and the red horizontal line, respectively. On each box, the red mark shows the median, and the bottom and top edges indicate the 25th and 75th percentiles. The whiskers indicate the most extreme data points.

The DEL10 from the measured moment $T_{m,tilt}$ are now compared to the results from the simulated $T_{s,tilt}$. Fig. 2 shows the probability density function (PDF) of the DEL10 of the measured $T_{m,tilt}$ for the same wind bin of $u = 8 \pm 0.5$ m/s and TI $= 10 \pm 1$% investigated before for the simulated data. A set of 149 time series of $T_{m,tilt}$ from the full-scale Nordex turbine belongs to this bin. The green and red lines in Fig. 1, depicting the maximum and minimum DEL10 from the simulated $T_{s,tilt}$, are now vertically placed over the PDF in Fig. 2. As observed, the PDF of the DEL10 from the measured $T_{m,tilt}$ spread out further than the simulated $T_{s,tilt}$. This especially holds for the larger values of the DEL10, depicted by the right tail of the distribution.

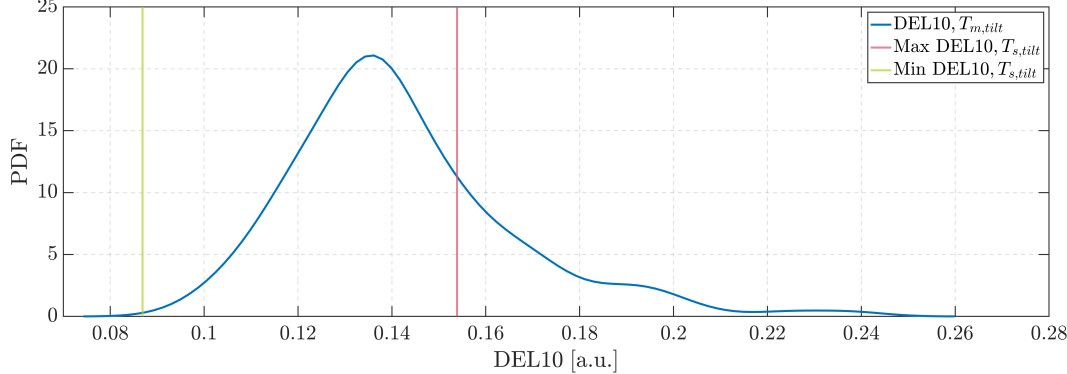

**Figure 2.** Probability density function (PDF) of normalized DEL10 for the measured tilt moment ($T_{m,tilt}$) of the full-scale operating Nordex wind turbine for the wind bin $u = 8 \pm 0.5$ m/s and TI= $10 \pm 1$%. DEL10 minimal and maximal values from simulated $T_{s,tilt}$ are marked by the green and red vertical lines (shown horizontally in Fig. 1).





As mentioned previously, the selection of the exponent $m = 10$ was intended to focus on the large-amplitude events of the load signal for the calculation of the DELs (see Eq. (1)). It is acknowledged that atypical large-amplitude loads may induce undesirable effects on the turbine, and there is a clear necessity for manufacturers and operators of wind turbines to enhance their understanding of such effects. At this point, we have demonstrated significant differences between the simulated and measured loads under comparable standard wind conditions. The discrepancies are observed when comparing the DEL10

values. By definition, large-amplitude events within the signal dominate the calculated DEL10 when a high value of the Wöhler exponent, e. g., $m = 10$, is used as explained in Sec. 2.1. The next step is to identify and investigate the characteristics of the large-amplitude load events, which, as shown in Fig. 2, are measured in operating full-scale circumstances but underestimated by the numerical simulations.

## 4    What is dominating the Damage Equivalent Loads?

Having identified the differences in the DEL10 between simulated and measured loads, we now investigate the origin of the large DEL10 values within the measured data. For doing so, we analyze the time series of the measured $T_{m,tilt}$ and $T_{m,yaw}$ whose DEL10 values exceed the maximal DEL10 values of the corresponding simulated $T_{s,tilt}$ and $T_{s,yaw}$ .

Interestingly, we found within those selected time series of $T_{m,tilt}$ and $T_{m,yaw}$ load events that appear as isolated 'bumps'. An example of such a bump is presented in Fig. 3. The bump structure lies inside the shadowed interval. The typical time scale

of those bump events ranges from $20\,\mathrm{s}$ to $50\,\mathrm{s}$ and tends to shorten with higher wind speeds. Automated detection of the bump structures can be implemented by fixing an amplitude threshold to the peaks of a filtered version of the signal.

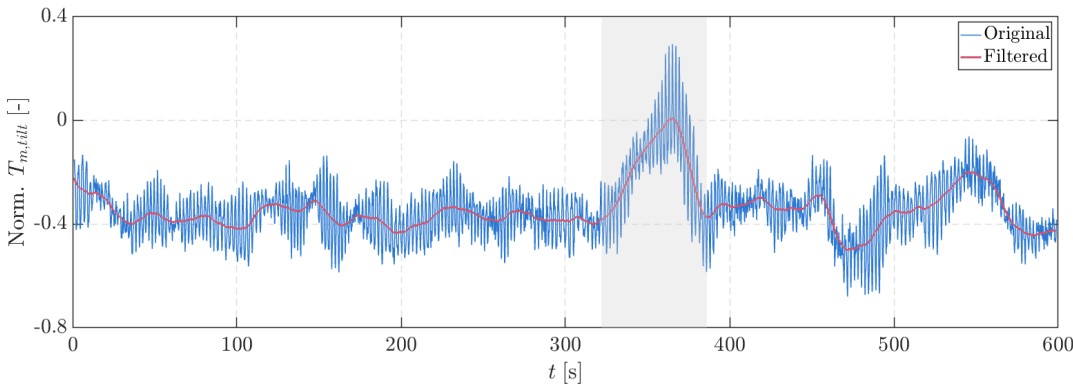

**Figure 3.** Normalized 10-min time series of the measured tilt moment ($T_{m,tilt}$ ) at the full-scale Nordex turbine with a DEL10= 0.22, and an observable bump event enclosed by the shadowed area. The time series belongs to the wind bin $\bar{u} = 8\,\mathrm{m/s}$, and TI = 10%. The blue line shows the original $20\,\mathrm{Hz}$ signal. The red line depicts a low pass filtered version of the signal with a cutoff frequency of $0.1\,\mathrm{Hz}$.

Then, the questions arise: what is the contribution of such bumps to the DEL10? Are such bumps the main drivers of the calculated DEL10? According to Eq. (1), the calculation of the DEL10 is defined as the aggregate of the amplitudes $s_i$ within the signal weighted to the power of $m$, i. e., $\mathrm{DEL10} \propto \sum n_i s_i^{m=10}$. Therefore, the largest amplitude $s_i$ within the time series





dominates the DEL10. As it will be proved, the global maximal and minimal values within the load signal delineate this largest amplitude driving the DEL10.

  To demonstrate this, we take the exemplary 10-min time series of the measured $T_{m,tilt}$, shown in Fig. 3. Two artificial load signals with the same global maximum and minimum are generated and compared. The artificial signals are generated using two components:

i) a high-frequency contribution characterized by a 3P-frequency (i. e., three times the rotational frequency of the main shaft), with an amplitude which equals the mean amplitude of the 3P frequency within the measured $T_{m,tilt}$ (calculated e. g. by FFT),

  ii) a low-frequency contribution with the mean equal to the mean of the measured $T_{m,tilt}$, and additional smooth bumps generated using fifth-order polynomials.

The comparison of $T_{m,tilt}$ to the two artificial signals is shown in Fig. 4. In a) the artificial signal contains only the largest low-frequency positive bump, i. e., over the mean. In b) the artificial signal contains both, the largest positive and the largest negative, i. e., below the mean, low-frequency bumps. In b), the global minimum and maximum of the $T_{m,tilt}$ and the artificial signals are equivalent, with values of -0.68 and 0.29, respectively. The calculated DEL10 are given in the legends of the plots. As shown in Fig. 4, if the minimum and maximum of the measured and artificial signals do coincide, their DEL10 will almost

be the same. In the example in b), the values of the DEL10 differ by 3% between the $T_{m,tilt}$ and the artificial.

  Further analysis of multiple signals of the measured $T_{m,tilt}$ and $T_{m,yaw}$ suggested that the global maximum and minimum, which drive the DEL10, are delineated by two characteristics of the load signal: the amplitudes of the high-frequency content and the dynamics of the low-frequency component. The high-frequency contribution of the tilt and yaw moment at the main shaft is mainly induced by the rotation of the three blades as they pass through turbulent eddies (Burton , 2011). The low-

frequency part, however, is driven directly by changes in the incident flow in the rotor plane.

  To isolate the low-frequency contribution, we apply a Butterworth low-pass filter (Butterworth , 1930) to the load signals. The cutoff frequency is $0.1\,\mathrm{Hz}$. This value is lower than the 1P frequency, i.e, rotational frequency, of the turbine. The filter is applied forward and backward to the signals so as not to run into time shifts. A filtered version of a load signal was already shown in Fig. 3 for a 10-min time series of the measured $T_{m,tilt}$. In Sec. 5.2 we will resume the discussion about the filtered

signals of the bending moments and their correlation to structures in the incoming wind.

## 5 The virtual Center of Wind Pressure

In the previous section, we highlighted the fact that bump events in the corresponding load time series dominate the Damage Equivalent Loads. In the present section, we aim to relate such bump events to specific characteristics of the wind field itself. Further investigation of the measured data from the Nordex turbine (i. e., of additional load sensors) showed that significant

changes in the bending moments at the main shaft coincide with specific events on the bending moments of the individual blades. These events in the loads on the blades were found to be correlated with specific azimuthal sections within a revolution.



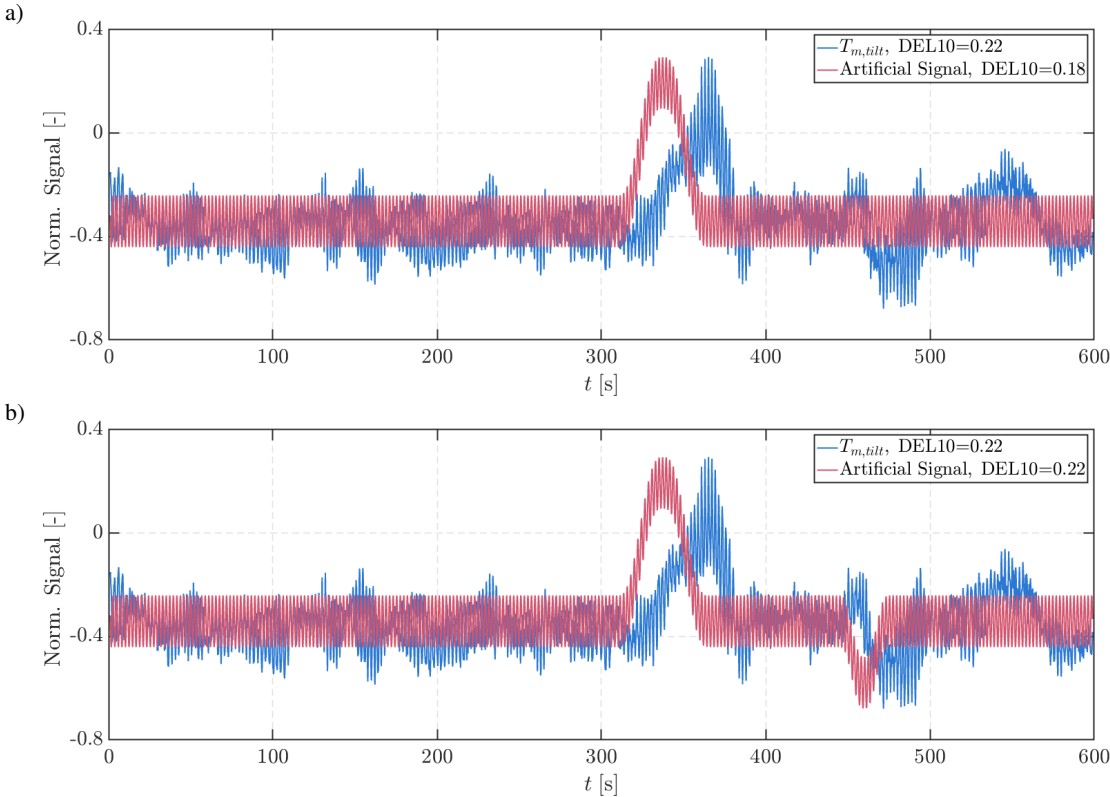

**Figure 4.** Comparison of the normalized measured signal $T_{m,tilt}$ at the full-scale Nordex turbine (blue line) and two artificial signals (red lines). a) the artificial signal has only the largest positive low-frequency bump between $300\,\text{s}$ and $400\,\text{s}$. b) the largest negative low-frequency bump is also added to the artificial signal at around $450\,\text{s}$. The resulting DEL10 are given in the legends of the plots. 'Positive' and 'negative' bumps refer to structures over and below the mean value.

This observation led to the formulation of a hypothesis that wind structures appearing exclusively in specific regions of the rotor plane might explain the occurrence of the load bumps in the bending moments, i. e., $T_{m,tilt}$ and $T_{m,yaw}$ , at the main shaft.

The discussion concerning the origin of these unusual bump events on the measured loads has prompted the need to obtain information not only on the wind speed at different heights, as is the case of the met-mast data provided at the site of the Nordex turbine (see Sec. 2.3), but also on how the wind speed is distributed in the rotor plane. To this end, a study was conducted on GROWIAN wind measurements based on a double-met mast array. This configuration allows the investigation of spatial wind structures, not only in the conventional vertical direction but also in their horizontal dimension. More details on the GROWIAN

data were provided in Sec. 2.3.

The investigation of wind fields from the GROWIAN met mast array further supported the hypothesis that the persistent bend of the whole rotor and, thus, the main shaft must be related to a large wind structure, i. e., a change in the wind speed in





significant areas of the rotor plane. Further correlations between the main shaft bending moments and spatial wind structures will be discussed in Sec 5.2.

## 5.1 Definition

To quantify the impact of the assumed large-scale wind structures driving the bending of the man shaft, we use the well-known concept of *center of pressure* from fluid mechanics (Anderson, 1991). The center of pressure is defined as the point where the total sum of a pressure field acts on a body, causing a force to act on that point. Analogously, the total force acting on a body at the center of pressure is the surface integral of the pressure field across the body's surface. In a similar way, the effect of a wind field over the rotor of a turbine is reduced to a point-wise force acting at the *virtual center of wind pressure* (CoWP). As we outline below our definition of the CoWP will lead to a non-physical distance from the hub, this is why we refer to it as a virtual center. The concept is illustrated schematically in Fig. 5.

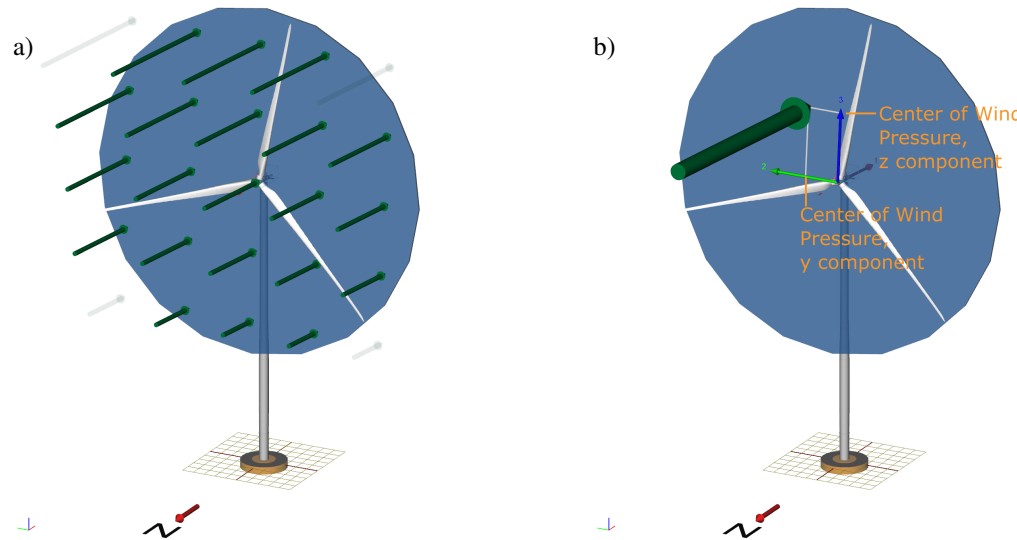

**Figure 5.** Schematic representation of the CoWP from a wind field acting on the rotor disk. a) Illustration of the discrete thrust forces acting on defined grid points ($y$-$z$). b) The collective effect of the individual thrust forces in a) is replaced by a single aggregated thrust force acting on the CoWP.

In the following, we will explain how to estimate the CoWP: As shown in Fig. 5, we define the surface $A$ of the rotor disk, i. e., given by the hub height and rotor radius as the surface of the body within the wind flow. This clarifies that, in the context of this paper, we assume the concept of dynamic pressure on an actuator disk (Sorensen, 2012). We can now calculate the yaw and tilt moments at the center of the disk ($o$), which are induced by the pressure on the surface. They will be named as the *virtual pressure-induced moments*. At this point, we introduce the subscript ($-v$) for virtual. Then, the two virtual moments



$T_{v,tilt}$ and $T_{v,yaw}$ are defined as,

$$T_{v,tilt}(t) = \int_A z \cdot \frac{\varrho}{2} \cdot C_T \cdot u^2(y,z,t)\, dA \tag{2}$$

$$T_{v,yaw}(t) = -\int_A y \cdot \frac{\varrho}{2} \cdot C_T \cdot u^2(y,z,t)\, dA. \tag{3}$$

where $z$ and $y$ are, respectively, the vertical and horizontal position within the rotor disk with origin $o$; $\varrho$ is the air density; $C_T$ is the thrust coefficient; and $u(y,z,t)$ is the normal component to $A$ of the wind field defined at the $y$-$z$ plane. We assume $C_T$ constant over the rotor disk and over time.

Now, the continuous definitions of the virtual pressure-induced moments are translated into their discrete versions as,

$$T_{v,tilt}(t) = \sum_{i=1}^{n} z_i \cdot \frac{\varrho}{2} \cdot C_T \cdot u^2(y_i,z_i,t) \cdot \Delta A_i \tag{4}$$

$$T_{v,yaw}(t) = -\sum_{i=1}^{n} y_i \cdot \frac{\varrho}{2} \cdot C_T \cdot u^2(y_i,z_i,t) \cdot \Delta A_i \tag{5}$$

where $n$ is the number of discretized grid points defined at $(y_i,z_i)$ which lie inside the rotor area $A$. $\Delta A_i$ is the discretized section of the rotor area $A$.

Generally, a moment $\boldsymbol{T}$ can be rewritten as the outer product of a point force $\boldsymbol{F}$, and a lever $\boldsymbol{r}$, as $\boldsymbol{T} = \boldsymbol{r} \times \boldsymbol{F}$. For the virtual pressure-induced moments, the force $\boldsymbol{F}$ is assumed as the thrust force $F_{thrust}$, normal to the rotor plane. The $F_{thrust}$ over the disk $A$ is calculated as,

$$F_{thrust}(t) = \sum_{i=1}^{n} \frac{\varrho}{2} \cdot C_T \cdot u^2(y_i,z_i,t) \cdot \Delta A_i. \tag{6}$$

Then, the $y$ and $z$ components of the lever $\boldsymbol{r}$ can be calculated from the $T_{v,tilt}$ and $T_{v,yaw}$, and $F_{thrust}$. This two-dimensional position at which the $F_{thrust}$ acts inside the rotor area $A$ corresponds to the CoWP. The two components of the CoWP, i.e., $CoWP_z$ and $CoWP_y$, are calculated as,

$$CoWP_z(t) = \frac{\sum_{i=1}^{n} z_i \cdot \frac{\varrho}{2} \cdot C_T \cdot u^2(y_i,z_i,t) \cdot \Delta A_i}{\sum_{i=1}^{n} \frac{\varrho}{2} \cdot C_T \cdot u^2(y_i,z_i,t) \cdot \Delta A_i} \qquad CoWP_y(t) = -\frac{\sum_{i=1}^{n} y_i \cdot \frac{\varrho}{2} \cdot C_T \cdot u^2(y_i,z_i,t) \cdot \Delta A_i}{\sum_{i=1}^{n} \frac{\varrho}{2} \cdot C_T \cdot u^2(y_i,z_i,t) \cdot \Delta A_i} \tag{7}$$

and,

$$T_{v,tilt}(t) = CoWP_z(t) \cdot F_{thrust}(t) \qquad T_{v,yaw}(t) = -CoWP_y(t) \cdot F_{thrust}(t). \tag{8}$$

The CoWP and the virtual moments ($T_{v,tilt}$ and $T_{v,yaw}$) can be calculated purely from measured or synthetic wind fields $u(y_i,z_i,t)$. In that way, the introduced concepts serve as tools for characterizing and comparing wind structures within different wind fields (e.g., at different atmospheric or orographic conditions or synthetic fields generated with different wind models). Furthermore, the surface area $A$ for calculating the CoWP can be adapted to different setups within experiments or measurement





configurations. For example, the FINO measurements are recorded at different heights by several anemometers vertically aligned (FINO). Therefore, in this case, the domain $A$ can be changed to a vertical line to characterize the one-dimensional

dynamics of the CoWP within the atmospheric inflow.

## 5.2    Correlation between the Center of Wind Pressure and the bending moments

In this section, we investigate the correlation between the introduced CoWP, as a feature of the wind, and the induced bending moments at the main shaft of the wind turbine. The correlation between the CoWP and the bending moments is demonstrated using the GROWIAN wind measurements (see Sec. 2.3). To this end, the 334 blocks of 10-min wind measurements from the

double met-mast array were reconstructed as wind fields and applied within the alaska/Wind BEM simulator. Afterwards, the DEL10 from the signals of the tilt and yaw moments at the main shaft were calculated.

Interestingly, within the GROWIAN-simulated loads, we observed bump events on the signals, together with DEL10 values that exceed the DEL10 from IEC Kaimal-simulated loads when using comparable environmental conditions (i.e., $\bar{u}$ and TI). These findings agree with the results shown in Sec. 3, where large values of the DEL10 from the measured $T_{m,tilt}$ at the

full-scale operating turbine are not observed within the simulated $T_{s,tilt}$ with IEC Kaimal wind fields.

As we have now access to the full wind field, we are able to compare the virtual moments ($T_{v,tilt}$ and $T_{v,yaw}$), the CoWP, and the simulated moments ($T_{s,tilt}$ and $T_{s,yaw}$) from the GROWIAN data. The virtual moments $T_{v,tilt}$ and $T_{v,yaw}$ are calculated with Eqs. (4) and (5). Note that in the following, we refer to the simulated moments to those from GROWIAN reconstructed fields and not from standard Kaimal fields. In order to compare the three physically different signals, i.e., virtual

moments, the CoWP, and simulated moments, we normalize the data. The normalization allows a direct comparison of the dynamical behavior, and a meaningful cross-correlation. The normalization of the individual 10-min signals follows:

1. low-pass filtering of the data with a cut-off frequency of $0.1\,\text{Hz}$. By doing so, the 3P content of the signals is removed.

2. Subtraction of the mean value calculated over the 10-min length.

3. Normalization by the standard deviation calculated over the 10-min length.

Figure 6 and Fig. 7 show two exemplary 10-min periods of the three signals from the GROWIAN data. The wind conditions, i.e., $\bar{u}$ and TI differ between the two examples. As observed, the simulated $T_{s,tilt}$ correlates to the virtual $T_{v,tilt}$. Remarkably, the former is derived after the interaction of the wind and the turbine, while the latter is calculated entirely from the wind field. Furthermore, according to Eqs. (8), a correlation between the CoWP and $T_{v,tilt}$ was expected. However, the strong similarity seen in Fig. 6 and Fig. 7 reveals the dominance of the CoWP term over the thrust force $F_{\text{thrust}}$ on the estimated virtual moment

$T_{v,tilt}$. In other words, the low-frequency dynamics of the induced bending moment are driven by the CoWP's location rather than by the magnitude of the aggregated thrust force.



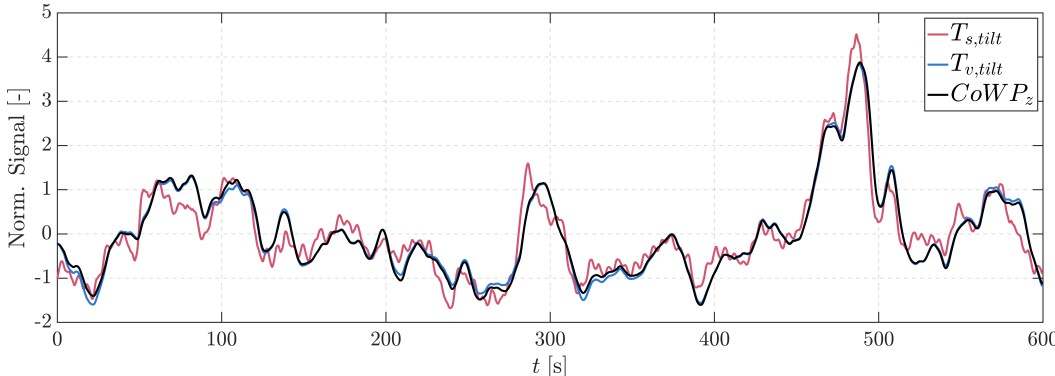

**Figure 6.** Comparison of the normalized signals: the simulated tilt moment ($T_{s,tilt}$), the virtual pressure-induced tilt moment ($T_{v,tilt}$), and center of wind pressure (CoWP). Data from a GROWIAN 10-min data set with $\bar{u} = 12.6$ m/s, and TI $= 7\%$ at 125 m height

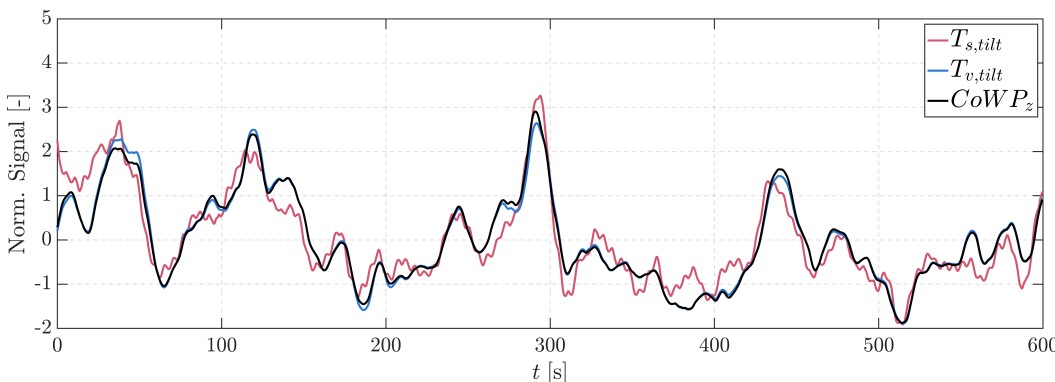

**Figure 7.** Comparison of the normalized signals: the simulated tilt moment ($T_{s,tilt}$), the virtual pressure-induced tilt moment ($T_{v,tilt}$), and center of wind pressure (CoWP). Data from a GROWIAN 10-min data set with $\bar{u} = 11.2$m/s, and TI $= 6\%$ at 125 m height.

We quantify the correlation between $T_{s,tilt}$ and $CoWP_z$. For that, the cross-correlation function $\rho(\tau)$ between the two quantities is calculated as,

$$\rho(\tau) = \frac{1}{\tilde{n}} \sum_{i=0}^{\tilde{n}} \left( CoWP_z(i) \cdot T_{s,tilt}(i+\tau) \right)$$

where $\tilde{n}$ is the number of time steps $i$ within the signal, and $\tau$ is the number of lagging steps. The values of $\rho(\tau)$ range between 0 and 1. Afterward, the maximal value $\rho_{max}$ of the cross-correlation function for each 10-min data set is calculated. Values of $\tau = [-20, 20]$ s are considered. Then, we define

$$\rho_{max} = \max_{\tau \in [-20, 20]} \left[ \rho(\tau) \right].$$





Fig. 8(a) shows on the $x$- and $y$- axis the wind conditions $\bar{u}$ and TI of each individual 10-min GROWIAN measurement at hub height (125 m). The color code shows the maximal value $\rho_{max}$ between the corresponding simulated $T_{s,tilt}$ and the CoWP. For a better visualization of the distribution of $\rho_{max}$ over the 334 GROWIAN data sets, the PDF of $\rho_{max}$ is shown in Fig. 8b.

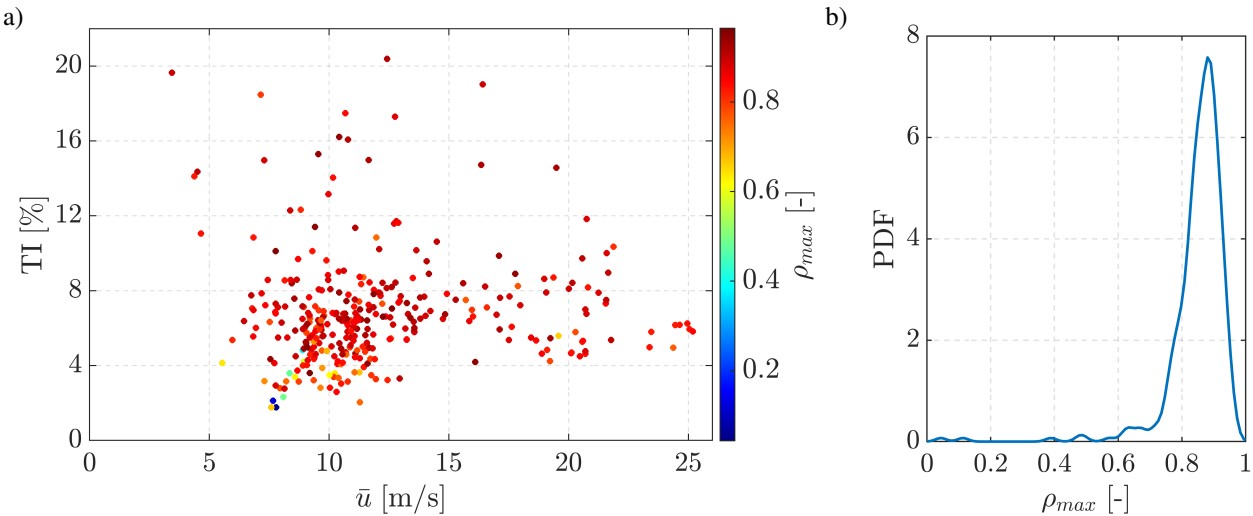

**Figure 8.** Maximum correlation coefficient $\rho_{max}$ between the CoWP and the simulated tilt moment ($T_{s,tilt}$) for the 334 GROWIAN measurements. In a) The position of the marker encodes the environmental conditions in terms of mean wind speed ($\bar{u}$) and turbulence intensity (TI). The color represents the value of $\rho_{max}$. In b) the PDF of $\rho_{max}$ is shown.

For relevant conditions, we showed that the CoWP captures the dominant contribution to the load dynamics, which is underlined by the correlation coefficients in Fig. 8. Except for low turbulence intensities and wind speeds, the correlation coefficient appears relatively robust ($\rho_{max} > 0.6$) concerning varying turbulence intensities and wind speeds. This includes even unstable atmospheric conditions corresponding to the cluster around 20 m/s due to strong westerly wind gusts as reported by Koerber (1988).

## 6   Conclusions and Outlook


We have shown that for modern wind turbines, there are certain situations where applying state-of-the-art synthetic turbulent wind fields within BEM simulations fails to fully reproduce the spectrum of measured loads for given environmental conditions, characterized by mean wind speed and turbulence intensity. The industry standard procedure for such uncovered situations is to superpose extreme operating gusts and turbulent wind fields. In order to increase the accuracy and efficiency in the design



process of wind turbines, and to generate site-specific turbulent wind fields and load estimates, it is essential to understand the source of these differences.

    We identified 'bump' events by analyzing the time series of measured bending moments at the main shaft, whose DEL values are particularly large. We demonstrated through the comparison to artificial signals that these bump structures drive the large DEL. The bumps were not observed within simulated loads from standard wind fields, which reinforces the need for a more
comprehensive understanding of the turbulent structures and the improvement of the synthetic wind fields.

    Using spatiotemporally measured wind speeds from the GROWIAN campaign, we have correlated those load bump events to large structures within the wind field. To describe those structures, we introduced the *virtual center of wind pressure* and *virtual pressure-induced moments*, which are obtained by a weighted average of the wind speed (more precisely, the thrust force) over the rotor disk. As the two quantities are independent of the turbine and are calculated from measurements or synthetic turbulent
wind fields, they are an efficient tool to characterize large-scale structures within the wind fields.

    The dynamic behavior of the virtual center of wind pressure strongly correlates with the main shaft moments of the wind turbine. In light of these results, we conclude that the current characterization of the inflow by single-point parameters, e. g., mean wind speed and turbulence intensity, do not account for events across the rotor plane and large-scale spatial structures within the wind that induce significant loads at the wind turbine. To close this gap, it would be desirable to extend the analysis
of the center of wind pressure to synthetic wind fields that account for the empirically observed occurrence of extreme wind field fluctuations, which are currently underestimated in the statistical framework provided by Mann or Kaimal models of the IEC standard. One attempt to better cover realistic wind field fluctuations was recently presented in Friedrich (2022).

    Ongoing studies of the virtual center of wind pressure using atmospheric measurement campaigns will help to parameterize its statistical and dynamic characteristics within the atmospheric flow. Synthetic turbulent wind fields, including realistic wind
features, will be essential for further applications, such as more accurate load predictions, optimized control strategies, and wind park optimization.

    A method for reconstructing signals of the loads at the main shaft based on the dynamics of the CoWP is presented in (Moreno, 2024a). Even though the high-frequency content of the signal is not reproduced by the method, due to the definition of the CoWP, the method proposes a fast approach for generating time series of the loads from the corresponding wind
field without directly relying on BEM simulations.

    In this work, only tilt and yaw moments at the main shaft of the wind turbine were investigated. The question of whether the concept can also be applied to other loads, e. g., blade or tower moments, remains open. An improved version of the concept could incorporate radial induction factors of the blades to give a weighted center of wind pressure. In that way, the effects of the wind structures on the individual blades might be better understood. Furthermore, it would be desirable to relate
the aggregated values of CoWP to other data analysis methods, e. g., local multifractal analysis of complex spatiotemporal random fields (Lengyel, 2022; Mukherjee, 2024) or statistical analysis of large-scale wind field structures (Moreno, 2024b). For the design process of turbines as recommended by the IEC, it could also be highly important to include the dynamics of the CoWP in synthetic wind fields using the methodology of *wind field constraints* (Dimitrov, 2017; Rinker, 2018; Friedrich, 2021), which could ultimately yield improved load estimations or control strategies, e. g., using Lidar measurements.





Further applications of the CoWP may also include the characterization of turbine wakes and thus help in wind park control and optimization.

*Code availability.*    The code for calculating the CoWP is based on basic programming functions. The authors might be directly contacted for further discussion and questions on the calculations.

*Data availability.*    An exemplary GROWIAN data set is provided. It corresponds to the results shown in Fig. 7.

*Author contributions.*    JS provided the field measurements, the simulation model of the full-scale wind turbine and raised the original question, CS performed the load calculations and correlation of load events with load sensors, DM performed analysis and correlations of turbulent inflow and load events. All authors contributed with their expertise in project meetings and discussions to understand and characterize the load phenomena which lead to the center of wind pressure, and also to the paper.

*Competing interests.*    Joachim Peinke is a member of the editorial board of Wind Energy Science.


*Acknowledgements.*    Financial support by German Federal Ministry for Economic Affairs and Energy within the scope of the project PASTA (03EE2024A/B/C) is gratefully acknowledged.





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
