# Peer review of "Introduction of the Virtual Center of Wind Pressure for correlating large-scale turbulent structures and wind turbine loads"

_Wind Energy Science, 2025_

## Referee Comment (RC2)

**Review: Introduction of the Virtual Center of Wind Pressure for correlating large-scale turbulent structures and wind turbine loads**

**General Comments**

The study considers two sets of observations from vertical towers in the presence of wind turbines to compare the loads generated from observations versus simulated wind fields generated using the IEC standards. The paper argues that "bumps" in the loads are the major contributors to the Damage Equivalent Load (DEL) and that these bumps are not found in the loads produced from the simulated wind field (not shown). The only analysis of the wind field is a calculation of the virtual center of wind pressure and it is shown that this metric correlates well with the simulated tilt moments.

Overall, the writing quality of this paper is good, but there remains a glaring omission from the paper that the actual wind field is not shown in any way. The calculated and simulated loads are compared, but the data with which they were calculated is omitted. It is difficult to believe in any of the findings when the wind field comparisons are missing. Due to this and other major revisions listed below, I recommend the paper be reconsidered after major revisions are completed.

**Major Revisions**

• The paper compares the calculated loads which are a function of either the observed or simulated wind fields, but doesn't show results from the actual wind fields. Without seeing how these wind fields compare, it's tough to believe that the differences in the loads are not caused by some major discrepancy in the wind fields. By showing that the simulated wind field is reasonably close to that of the observed winds, readers can be more confident in trusting the analysis. If the modeled wind field is not a time series, but just a PDF, for example, then a comparison of PDFs would be fine. Additionally, discussion of the differences between the simulated and observed wind fields would allow for the authors to generate hypotheses about what the differences in the loads may look like, and/or attribute the differences in the loads to aspects of the differences between simulated and observed wind fields more clearly.

One of your conclusions states, "[t]he bumps were not observed within simulated loads from standard wind fields, which reinforces the need for a more comprehensive understanding of the turbulent structures and the improvement of the synthetic wind fields," I don't see where in the paper this is shown. If the "bumps" are inherently not in the time series due to the simulated wind field being a statistically-generated field, then explain that (and, show the simulated wind field that was used to create the time series).

- Section 4 what is the DEL when there is no "bump" in the artificial signal? It's argued that these "bumps" are the main drivers of DEL but there is nothing showing the DEL before the bump is introduced.
- Section 5.2 it is lost on me which data are being used to calculate these variables (Ts,tilt, Tv,tilt, Tm,tilt, etc.). It seems like from the text the "simulated" moments are from the observed GROWIAN data, the "virtual" moments are using an equation (but is it from simulated data or observed?), but then the "measured" moments are never shown. This section is very unorganized and difficult to follow. Additionally, the lack of clearly showing a direct comparison between observations and simulations truly hurts the credibility of the research.
- I don't understand the "bump" event correlation to "large structures" in the wind field. The load is a function of the wind field; so if there is a signal in the load, there is a signal from the wind field, no? This, again, would benefit from showing the wind field.

**Minor Revisions**

 In Figures 1 and 2 (possibly others), you show data for either the simulated or observed wind field, but not both at the same time. Is it not possible to plot the same metrics for both simulated and observed? If you want to drive home the comparison, it seems like it would be beneficial to plot both of these together to convince the reader that the simulated data is or isn't comparable to the observed data.

**Technical Suggestions**

• Writing and figure quality are good - no suggestions.

---

## Author Comment (AC2)

**Reply to Referee Comments**

**Introduction of the Virtual Center of Wind Pressure for correlating large-scale turbulent structures and wind turbine loads**

Referee's comment (RC) in blue

Author's comment (AC) in black

**General summary:**

We would like to thank the referees for their constructive comments and inspiring questions. We followed most of the comments and questions which substantially improved our manuscript.

We did not follow two suggestions:

- We did not add LES wind fields as recommended by referee 1. We fully agree that this would be an interesting expansion. The fundamental question underlying our work arose in industrial application and was investigated using process chains prescribed by certification guidelines. At the moment, this leaves out LES wind fields. Additionally, using LES fields does pose many additional questions. How to couple LES to the aeroelastic simulation: Using BEM, actuator disk or actuator line? And how does this alter the *center of wind pressure*? There will be follow-up works investigating how LES wind fields behave in this situation and if they can improve the simulation outputs compared to the synthetic standard wind fields.

- We could not follow the suggestion made by referee 2 to compare the metric *center of wind pressure* from field measurements to the measured tilt moment. This is a highly desirable step, but unfortunately cannot be carried out with the available data. The simple reason for this is that spatially distributed wind measurements are required to calculate the *center of wind pressure* and derived characteristics. These are not available from the field measurements (as is the case with most measurement campaigns). The new *Research Wind Farm WiValdi* from DLR could provide suitable measurements for this comparison.

Below, we respond in detail to all of the reviewers' comments and refer to the line numbers in the revised version of the manuscript. The revised manuscript and a diff file highlighting the changes between the original and the revised manuscripts are provided along with this document.

**Referee Comment 1**

The present work proposes an additional parameter for describing wind fields in order to better understand differences between measured and simulated loads. The concept of the virtual center of pressure - which in some ways complements the rotor-averaged wind speed by providing spatial information- could be a useful tool for comparing i.e. measured wind fields, LES wind fields, and synthetic wind fields (as exemplified here). The inclusion of LES wind fields would have been a very useful addition to the work, as this would also provide insight into whether such wind fields could be used to improve design analysis.

We agree with the referee and want to amend: The original question of our research was raised by industry, and the focus of the underlying research project was to check and improve methods which have to be used in industrial guidelines (IEC) for certification, and which are numerically efficient enough to be used in design processes. This leaves out LES fields – at the moment, also as for LES investigations, many other questions arise, whether an actuator disk, an actuator line, or a blade resolved simulation can be used. And, how does this alter the CoWP. There will be follow-up work and articles which include LES wind fields.
We thank the referee for the comment and sharpened the focus of our work as well as the above comments to other wind fields within the manuscript – starting with line 119.

The argumentation for the virtual center of pressure is based on assuming a constant thrust coefficient over the rotor, which is not a very convincing argument. I don't see any real reason for introducing the thrust at all, as the formulation is really just a spatial averaging of the longitudinal wind speed squared.

The center of wind pressure is the position where the spatial average of the wind speed squared is valid. The thrust coefficient is added to get a physical interpretation of the new characteristic property. Thus, it is possible to get an impression of the property and draw instructional sketches as in figure 5. Otherwise, it will be a measure without any physical representation.

It is correct that the thrust coefficient in the equation is not necessary – as it cancels out. But, for a better understanding and further potential enhancements of the measure, it is added. We explained that more thoroughly in the preprint and gave hints on how to vary the constant thrust coefficient for other applications in the future –starting at line 268.

The use of DEL10 to identify large load cycles also seems a bit contrived to me. The components of interest are generally made of steel, and damage would typically be calculated with a much lower exponent. For the bearings, different damage formulations are typically used (i.e. load distribution duration). The argument for using DEL10 is that this metric emphasizes large load ranges. Why not simply examine the distribution of load ranges? If using DEL, there should also be more discussion of the counting algorithm that is used, particularly how unclosed cycles are addressed (since large load cycles can remain unclosed at the end of the time series). Why is it not possible to show a distribution based on simulations in Figure 2? This would be better to compare against measurements.

We agree on many aspects. The DEL with exponent m = 10 is not a common choice for loads at the main bearing. We mainly used the DEL10 as a tool to identify those uncommon large load

events. This could have also been done by studying load distributions as proposed – we chose to use the DEL as a tool for the following reasons:

- It is a common tool available in each post-processing chain
- It yields a scalar number which can easily be evaluated for each load situation (in our case, for each wind bin) and can be much easier judged as time series and distributions

We explained the usage of the DEL as a tool for detecting and evaluating uncommon load situations and the used exponent more clearly in the preprint – starting at line 137.

We also thank the referee for the suggestion to add the distribution of calculated loads (from figure 1) in the same manner as for the measured ones in figure 2 of the preprint. We changed figure 2 appropriately, which improves our argumentation.

For rotor-level bending moments, it seems rather obvious that these should be dependent on the spatial variation of the wind field rather than the magnitude of the thrust itself. Other authors have proposed an asymmetry index (i.e. Lavely's PhD thesis) that could also be a useful metric for comparison to the proposed metric.

We thank the referee for the hint and the related work. We added remarks on how to adapt the introduced metric for other load signals and added the metric from the cited thesis as another example starting at line 268.

**Referee Comment 2**

**General Comments**

The study considers two sets of observations from vertical towers in the presence of wind turbines to compare the loads generated from observations versus simulated wind fields generated using the IEC standards. The paper argues that "bumps" in the loads are the major contributors to the Damage Equivalent Load (DEL) and that these bumps are not found in the loads produced from the simulated wind field (not shown). The only analysis of the wind field is a calculation of the virtual center of wind pressure and it is shown that this metric correlates well with the simulated tilt moments.

Overall, the writing quality of this paper is good, but there remains a glaring omission from the paper that the actual wind field is not shown in any way. The calculated and simulated loads are compared, but the data with which they were calculated is omitted. It is difficult to believe in any of the findings when the wind field comparisons are missing. Due to this and other major revisions listed below, I recommend the paper be reconsidered after major revisions are completed.

**Major Revisions**

The paper compares the calculated loads which are a function of either the observed or simulated wind fields, but doesn't show results from the actual wind fields. Without seeing how these wind fields compare, it's tough to believe that the differences in the loads are not caused by some major discrepancy in the wind fields. By showing that the simulated wind field is reasonably close to that of the observed winds, readers can be more confident in trusting the analysis. If the modeled wind field is not a time series, but just a PDF, for example, then a comparison of PDFs would be fine. Additionally, discussion of the differences between the simulated and observed wind fields would allow for the authors to generate hypotheses about what the differences in the loads may look like, and/or attribute the differences in the loads to aspects of the differences between simulated and observed wind fields more clearly.

We agree with the referee that a more thorough explanation of the synthetic wind fields is needed. We explained the methods to generate the synthetic wind fields and information on how they compare to observed/measured wind fields in two sections in the added appendix starting at line 373.

We also explained that comparing wind fields from the field measurements with the full-scale turbine is not possible, as the met mast captured only three different vertically aligned points up to the hub height of the turbine. To capture the full dynamics of the turbine, a spatial wind measurement is necessary. This is the reason why we used the GROWIAN measurements and applied them within the simulation. In the added appendix B (starting at line 385), we compared extreme GROWIAN measurements statistically with a reasonable number of synthetic wind fields, which are generated using the characteristics of the measured GROWIAN field (i.e. mean wind speed at hub height and turbulence intensity). This shows clearly, that observed and

synthetic wind fields are comparable in terms of turbulence intensity and mean wind speed at hub height. Additionally, it shows that the observed wind fields exhibits extreme behavior – showing as bump-like structures – within the newly introduced virtual center of wind pressure, which does correlate to the main bearing moments.

One of your conclusions states, "[t]he bumps were not observed within simulated loads from standard wind fields, which reinforces the need for a more comprehensive understanding of the turbulent structures and the improvement of the synthetic wind fields," I don't see where in the paper this is shown. If the "bumps" are inherently not in the time series due to the simulated wind field being a statistically-generated field, then explain that (and, show the simulated wind field that was used to create the time series).

We added appendix B to show the suggested comparison. Here, we show a GROWIAN wind field generating the load bump. Statistical properties of this wind field were compared to standard synthetic wind fields, which show, that the mean wind speeds are comparable. However, the "virtual center of wind pressure" of the GRWOWIAN measurement exhibits a bump- which correlates with the load bump – that cannot be compared to any of the standard wind fields. Appendix B starts at line 385. See especially Figure B3 for a comparison of the wind fields.

Section 4- what is the DEL when there is no "bump" in the artificial signal? It's argued that these "bumps" are the main drivers of DEL but there is nothing showing the DEL before the bump is introduced.

We totally agree and thank the referee for the proposal. We added a figure (new Figure 4 a)) with an artificial signal without any bump and the corresponding DEL10 value – which improves our argument.

Section 5.2- it is lost on me which data are being used to calculate these variables (Ts,tilt, Tv,tilt, Tm,tilt, etc.). It seems like from the text the "simulated" moments are from the observed GROWIAN data, the "virtual" moments are using an equation (but is it from simulated data or observed?), but then the "measured" moments are never shown. This section is very unorganized and difficult to follow. Additionally, the lack of clearly showing a direct comparison between observations and simulations truly hurts the credibility of the research.

We would also like to provide a comparison between observed and simulated data.

Measured moments (from the real turbine) only exist for Nordex field measurements. For those, the wind speed was recorded at only three vertically aligned points. Therefore, there is not sufficient data for calculating the virtual moments from spatially resolved wind fields. Thus, virtual moments cannot be compared to measured ones. We explained that within the manuscript starting at line 274.

To be able to compare measured moments, virtual moments, and "simulated moments", there must be simultaneous measurements from spatial wind field and turbine loads. Those data may be captured within the new wind research farm WiValDi, but are not available to the authors.

We agree that section 5.2 requires further explanation and summarization of variables.

In an additional list, we explain the variables that are evaluated and correlated in this section (see lines starting at line 291). For each variable, we specify the source (data or simulation) and, if applicable, the formula used, which improves the readability of our manuscript.

I don't understand the "bump" event correlation to "large structures" in the wind field. The load is a function of the wind field; so if there is a signal in the load, there is a signal from the wind field, no? This, again, would benefit from showing the wind field.

"if there is a signal in the load, there is a signal from the wind field". This is exactly the question we wanted to address with this preprint. Here, the specification of the question is: "Which" signal from the wind field can be used to identify those load signals or events? With the load statistics in section 3 we have motivated, that the mean wind speed at hub height and turbulence intensity are not sufficient. We introduced the center of wind pressure as a new metric to identify those load events.

As suggested by the referee, we added plots from a measured GRWOWAN wind field and 32 synthetic wind fields generated following the IEC standards for the same environmental conditions (mean wind speed and turbulence intensity) within Appendix B. The new figures (especially Figure B3) show that the mean wind speed over the whole domain or at hub height is not sufficient to detect extreme load situations and demonstrate how center of wind pressure can be used to display those events.

**Minor Revisions**

In Figures 1 and 2 (possibly others), you show data for either the simulated or observed wind field, but not both at the same time. Is it not possible to plot the same metrics for both simulated and observed? If you want to drive home the comparison, it seems like it would be beneficial to plot both of these together to convince the reader that the simulated data is or isn't comparable to the observed data.

We thank the referee for the hint and changed Figure 2 for statistically observed and simulated data from the same wind bin – which improves our argumentation.

For special events measured in the field turbine, comparisons of a singular time series to a singular simulation are not possible, as we only have three measurement locations of the wind speed. There is no measured wind field, that can be applied within the simulation. So, the comparisons of the GROWIAN fields to comparable Kaimal fields from the same wind bin are the best available approach.

**Technical Suggestions**

Writing and figure quality are good- no suggestions.